# Nano-Mechanical Properties of Surface Layers of Polyethylene Modified by Irradiation

**DOI:** 10.3390/ma13040929

**Published:** 2020-02-19

**Authors:** Martin Ovsik, Miroslav Manas, Michal Stanek, Adam Dockal, Ales Mizera, Petr Fluxa, Martin Bednarik, Milan Adamek

**Affiliations:** 1Faculty of Technology, Tomas Bata University in Zlin, Vavreckova 275, 760 01 Zlín, Czech Republic; stanek@utb.cz (M.S.); a_dockal@utb.cz (A.D.); fluxa@utb.cz (P.F.); mbednarik@utb.cz (M.B.); 2Faculty of Applied Informatics, Tomas Bata University in Zlin, CEBIA-Tech, Nad Stranemi 4511, 760 05 Zlín, Czech Republic; manas@utb.cz (M.M.); mizera@utb.cz (A.M.); adamek@utb.cz (M.A.)

**Keywords:** polymers, crosslinking, electron rays, nano-indentation, X-ray, gel content, spectroscopy

## Abstract

This study’s goal was to describe the influence of a wide range of ionizing beta radiation upon the changes in surface layer mechanical properties and structural modifications of selected types of polymer. Radiation crosslinking is a process whereby the impingement of high-energy electrons adjusts test sample structures, thus enhancing the useful properties of the material, e.g., hardness, wear-resistance, and creep, in order that they may function properly during their technical use. The selected polymers tested were polyolefin polymers like polyethylene (Low-density polyethylene LDPE, High-density polyethylene HDPE). These samples underwent exposure to electron radiation of differing dosages (33, 66, 99, 132, 165, and 198 kGy). After the crosslinking process, the samples underwent testing of the nano-mechanical properties of their surface layers. This was done by means of a state-of-the-art indentation technique, i.e., depth-sensing indentation (DSI), which detects the immediate change in the indentation depth associated with the applied force. Indeed, the results indicated that the optimal radiation dosage increased the mechanical properties by up to 57%; however, the beneficial levels of radiation varied with each material. Furthermore, these modifications faced examination from the structural perspective. For this purpose, a gel test, Raman spectroscopy, and crystalline portion determination by X-ray all confirmed the assumed trends.

## 1. Introduction

The research papers and literature cited in this study have focused on description of the influence of ionizing radiation on the tested polymers’ properties. These sources are, however, insufficient, since they do not contain the required amount of complex information. In addition, relevant information about the correlation between ionizing beta radiation and morphology development is also lacking. Unfortunately, this means that the exact modification of a polymer’s mechanical properties by radiation crosslinking for industrial applications is quite challenging.

The chemical linking of two polymer functional groups produces linear polymers. Furthermore, if one of the groups is multifunctional, swelling and elongation with branching is observable. This leads to the creation of an infinite 3D network (Figure 1), i.e., a gel.

In all cases, there are crosslinking actions that occur sequentially [1]:Dimensions and poly-dispersion grow in the first phase;As the reaction reaches gel point, the molar weight increases and the gel starts to appear in the system;The system includes two parts—on the one hand, an infinite structure called “gel”, on the other, a cluster of molecules with finite size called “sol”. Extraction of molecules leads to its separation from the gel;It is not possible to extract the complete network, byproducts, and micro-gel. The gel is insoluble and, if exposed to a solvent, it experiences imbibition;Finally, the sol content, as well as its molar weight and poly-dispersity, decreases as the reaction continues.The gel network creates active elastic chains. These chains carry the applied stress, thus determining the value of the gel elastic modulus and its imbibition equilibrium degree.

This study dealt with the nano-mechanical properties of two polymers, namely low-density polyethylene (LDPE) and high-density polyethylene (HDPE), after their modification by ionizing radiation in comparison to their nano-mechanical properties prior to irradiation. The first polymer considered in this research was HDPE. Polyethylenes are commodity polymers, and they account for around 70% of total plastics consumption. They are widely available, low-cost, and easy to process. Typical applications for their use include medical applications, water pipes, the packaging industry, etc. HDPE is a semi-crystalline linear thermoplastics, commonly used to produce electrical insulation, shrinkable products, tubing, packaging, etc. [3]. Cassidy et al. dealt with the irradiation of HDPE using gamma radiation. They investigated the increase in strength, although the material brittle after irradiation up to 60 kGy—and simultaneously, the increase of the ultimate strength, the decrease of elongation, and the increase of Rockwell hardness [4]. Other research [5] has concentrated on gamma irradiation of HDPE with irradiation doses up to 475 kGy. In this case, the gamma radiation improved the biocompatibility parameters of HDPE, such as surface free energy, adhesion work, Good–Girifalco’s parameter, wettability, and surface roughness. These improvements were caused by the combination of oxygen and free radicals formed after irradiation that led to the increase of the surface polarity [6]. The connection of two or more different materials, or their reinforcement, offers a new spectrum of material properties. The reinforcement of HDPE with hydroxyapatite and its subsequent irradiation by an electron beam showed that the improvement of material toughness is due to radiation crosslinking [7]. 

Wang and others [8,9,10] investigated changes in the mechanical properties of PP and PP/HDPE blends caused by irradiation. They found out that after irradiation by gamma radiation of PP/HDPE blends, the impact strength increased. The higher the content of HDPE, the smaller the impact strength. 

Two main types of radiation, ionic and ionizing, are used to modify polymers. Owing to its properties, beta radiation (electrons) is widely used. The emitted electrons accelerate through an electrostatic field between a cathode and an anode. The penetration depth increases with the electrons’ energy. In addition, the penetration depth depends on the density of the irradiated material, and the geometry of the product [11,12,13,14,15,16]. Most previous studies [17,18,19,20,21,22,23] examining the effect of irradiation on the resulting properties of materials have primarily focused on describing the structural changes, mechanical properties, and chemical and temperature resistance dependence on radiation doses. Gheysari et al. [5] demonstrated the positive effect of high-energy beta radiation on the mechanical and thermal properties of LDPE and HDPE, respectively, which were modified using radiation doses ranging from 50 to 250 kGy.

The effect of radiation on the properties of polymer materials have been addressed in a number of presented works. However, there is not enough detailed information on the mechanical properties of radiation crosslinked polymers, especially in the micro- and nano-fields. The paper therefore aimed to describe changes in the surface properties of commercial polyethylenes exposed to accelerated electrons with radiation doses ranging from 33 to 196 kGy.

Depth-sensing indentation (DSI), a modern method, was used to measure the selected polymers’ nano-mechanical properties—indentation hardness, modulus, and creep, for example. The results helped to determine the optimal radiation dosage for each of the tested materials. Moreover, gel content measurements confirmed the changes of nano-mechanical properties, as well as morphology studies using Raman spectroscopy and X-ray.

## 2. Materials and Methods

Polymer surface layer studies are wide-ranging, including—for example, nano-mechanical properties or structural changes examinations. This research dealt, in particular, with the study of nano-mechanical property modifications in LDPE and HDPE induced by radiation crosslinking.

Ten statistical correctness measurements were used to test all of the mentioned properties. For the measurements, five sample were used.

### 2.1. Material

Selection of a suitable material depends on the effective properties required of the final product. Specifically, one desired effect is to add some properties of high-tech polymers into standard and construction polymers by crosslinking, which can be achieved by irradiation, thus replacing expensive materials in many applications.

This study also investigated polyolefin polymer materials. LDPE and HDPE are the most commonly used polymers, as shown in Figure 2, in the form of a polymer pyramid. Materials that can be crosslinked are marked in red.

To measure the changes of the material properties after the exposure to radiation crosslinking, polyethylene was chosen. This choice was made due to ability of this polymer to crosslink without adding any poly functional monomers (crosslinking agents) to the structure (polymer matrix). The crosslinking agents are generally added to polymer materials in order to increase the effectivity of the crosslinking process. Table 1 shows the tested polymers, which were divided as follows.

### 2.2. Sample Preparation

Test sample preparation made use of injection molding on ARBURG Allrounder 470e machines (Loßburg, Germany). Process parameters (displayed in Table 2) were set to the manufacturer’s recommendations. In accordance with supplier guidelines, the selected materials were pre-dried (the temperature of drying was 60 °C for the duration of 1 h) in an ARBURG THERMOLIFT 100-2 (Loßburg, Germany) drying device. Test specimens were produced in a bars shaped in agreement with the CSN EN ISO 179 standard. Figure 3 shows the test sample dimensions.

### 2.3. Irradiation

The samples underwent beta radiation under normal atmospheric conditions at room temperature (23 °C). The BGS Beta-Gamma-Service, GmbH & Co. The KG (Saal an der Donau, Germany) branch office performed this modification. The radiation source was a Rhodotron 10 MeV–200 kW (Tongeren, Belgium) toroidal electron beam accelerator. Dosage ranges were set in compliance with experience gained from industrial practice, to 33, 66, 99, 132, 165, and 198 kGy. Each cycle in the accelerator exposed the test sample to a radiation dose of 33 kGy. In addition, a dosimeter measured the absorbed radiation dosage, which was also subsequently determined using a Spectronic Genesys 5 (Goleta, CA, USA) photometric device.

### 2.4. Nano-Indentation Test

The irradiated polymer surface property measurement process made use of an (NHT^3^) nano-indentation tester manufactured by Anto Paar (Graz, Austria). The test complied with the CSN EN ISO 14577 standard. This technology is based on instrumented hardness tests, i.e., DSI principles that continually record the applied force “P” and the immediate location of the indentor “h”. A graphical representation of this dependence, describing the tested polymer’s behavior, is called an indentation curve. The Oliver and Pharr method was used to measure nano-mechanical properties and evaluation, e.g., indentation hardness, modulus, and creep.

The penetrating body was a Berkovich indentor (with an apex angle of 65.27°). Table 3 displays the process parameters used.

Indentation hardness (*H_IT_*, Figure 4) was calculated as the maximum load (*F*_max_) on the projected area of the hardness impression (*A*_p_) [26,27].
(1)$$H_{IT}=\;\frac{F_{max}}{A_{p}}$$
(2)$$A_{p}=23.96\cdot h_{c}^{2}$$

The indentation modulus (*E_IT_*) was calculated from the plane strain modulus (*E**) using an estimated Poisson’s ratio (*ν_s_*) sample (Polymer 0.3 to 0.4) [26,27,28].
(3)$$E_{IT}=E(1-{v_{s}}^{2})$$
(4)$$E^{*}=\frac{1}{\frac{1}{E_{r}}-\frac{1-v_{i}^{2}}{E_{i}}}$$
(5)$$E_{r}=\frac{\sqrt{\pi }}{2\cdot C\sqrt{A_{p}}}$$
where *E_i_* is the elastic modulus of the indenter (diamond 1141 GPa), *E_r_* is the reduced modulus of the indentation contact, and ν_i_ is the Poisson’s ratio of the indenter (0.07).

Determination of indentation creep *C_IT_* (where h_1_ is the indentation depth at time *t*_1_ of reaching the test force is constant) was done as follows; *h*_2_ is the indentation depth at time *t*_2_ of withstanding the constant test force [26].
(6)$$C_{IT}=\frac{h_{2}-h_{1}}{h_{1}}\times 100$$

### 2.5. Gel Content

A gel content test, performed in order to determine the insolvable gel content of the given material, was done in accordance with the ASTM D 2765 standard—Test Method C. A portion of 0.5 g weighed with a precision of five decimal places, was mixed with 100 mL of solvent on a “SWISS MADE EP 125 SM” weighing apparatus (Dietikon, Switzerland). Xylene was the solvent for the tested polymers, since it dissolved the amorphous part of the material, while leaving the crosslinked part intact. The mixture extraction duration was 24 h. To separate the solutes, these then underwent distillation. After removing the residual xylene, the crosslinked extract was dried in a vacuum for 8 h at 100 °C. The dried and cooled residue underwent reweighing, with a precision of five decimal places, and was compared to the original weight of the portion. The result, expressed as a percentage, is the degree of cross-linking.
(7)$$G_{i}=\;\frac{m_{3}-\;m_{1}}{m_{2}-\;m_{1}}\times 100$$
where *G_i_* is the degree of crosslinking of each specimen expressed in percentage; *m*_1_ is the weight of the cage and lid in milligrams; *m*_2_ is the total weight of the original specimen, cage, and lid in milligrams; and *m*_3_ is the total of the weight of the residue of specimen, cage, and lid in milligrams.

### 2.6. Infrared Spectroscopy

The relative numbers of hydroxyl and carbonyl groups on the surfaces of the given polymers were determined with the help of infrared spectroscopy. We used the attenuated total reflection (ATR) technique to measure the infrared spectra. A dry-air purged Nicolet 6700 FTIR spectrometer (Waltham, MA, USA) was used to measure the infrared spectra at the resolution of 2 cm^−1^, using 64 spectrum accumulations. The background was a pure ATR diamond crystal. OMNIC software 8.2 was then used to process the spectra, while ATR corrections were applied to modify them. The average of six measured spectra was used to determine the bond areas. Prior to averaging, the spectra—automatically adjusted to baseline—underwent normalization.

### 2.7. X-ray Diffraction

Crosslinking processes are prevalent in amorphous parts of polymers; therefore, the effectiveness of network creation decreases in line with increasing crystallinity. Crystallinity depends on multiple factors, including the linear chain structure, polar groups, secondary groups, and the degree of polymerization.

The X-ray diffraction patterns were determined using a PAN Analytical X-pert Prof X-ray diffraction system (Panalytical, Almelo, Netherlands). The CuK α radiation was Ni-filtered. The scans—(4.5° 2θ/min) were recorded in reflection mode within the range of 5–30° 2θ.

Each crystalline peak and amorphous halo had an individual Gauss function. Finally, the crystallinity of the samples was calculated as the ratio between the sum of the crystalline peak areas and the area of the whole X-ray pattern.

### 2.8. Raman Spectroscopy

An InVia Basis Raman Microscope manufactured by Renishaw (Gloucestershire, UK) was used to measure the Raman spectra). This device had a Leica DM 2500 confocal microscope (Boston, MA, USA) with a resolution of up to 2 M. The excitation source was a laser with a 514 nm wavelength and 20 mW output. We used 0.5 mW output, 10s measuring time, and 5 accumulation spectra for our test samples. The microstructure was recorded using a lens with 50× magnification. The track length was 2 M, and the test samples were measured in the range of 500 to 1600 cm^−1^.

## 3. Results

The first choice to be made in this study was of the materials suitable for radiation crosslinking in order to measure the nano-mechanical surface layer properties. The selection included a wide array of materials commonly used in technical practice, such as polyolefins (LDPE, HDPE). Technical practice frequently uses these materials due to their easy modification by beta radiation.

A DSI instrumented test was then done to measure the surface layer properties. The principle of this test is to detect the immediate depth of an indentation at an exact point in time. Finally, in order to confirm these results, we used structural measurement with the goal of describing changes in the structure induced by the electron irradiation. For the measurements, five samples were used. All of the observed properties were measured 10 times for each sample, and the resulting data were used to calculate the arithmetic mean and the standard deviation.

### 3.1. Nano-Mechanical Properties (Indentation Hardness, Indentation Modulus, Indentation Creep)

The main principle of the DSI method is the simultaneous detection of both the immediate changes of indentation depth and the increase/decrease of the applied load during the entire loading and de-loading process. The graphical representation of this test is an indentation curve, which displays the dependence of the indentation depth on force (Figure 5).

The first phase of the indentation cycle is a controlled stress application, during which the indentation device pushes against the test sample with a predefined force. The second phase of the cycle—denoted as de-loading—consists of the gradual decrease of the applied force all the way down to zero. Often, there is a delay between the aforementioned phases, during which the exposed sample faces the maximum force, thus allowing the measurement of the indentation creep.

Figure 6 displays the tested polymer properties and the dependence of the indentation depth on the indentation time before and after the application of varying radiation dosages. In Figure 6, individual depths (*h*_1_, *h*_2_) are presented. These variables were used to calculate the creep behavior. After reaching the maximum preset force (*F_max_*), the depth *h*_1_ was registered. At this moment, the indentor pressed into the test sample with a constant maximum force for the planned duration. Following the recording of *h*_2_, creep behavior was calculated. Figure 5 and Figure 6 show the noteworthy changes to these properties. These were recorded during their exposure to the individual radiation dosages.

The maximum indentation depth is an important parameter for the evaluation of the final nano-mechanical properties of the surface layer (Figure 7). The maximum depth for the tested LDPE measured by the indentation device tip was 12.9 µm. Every other specimen displayed a lower penetration depth. 

Indentation hardness (*H_IT_*) is the degree of material resistance to permanent deformation or damage. Figure 8 shows the evaluation of the indentation hardness and its dependence on the varying radiation dosages. These results show that radiation crosslinking increased the surface layer hardness of these materials. After the exposure to beta radiation, the test samples displayed an increase in indentation hardness. 

The highest indentation hardness values (26 MPa) were measured in LDPE samples irradiated with a radiation dosage of 132 kGy. The indentation hardness for the virgin material was 21 MPa. Irradiation of the 3D network led to a 23% increase of indentation hardness in comparison to the virgin material. However, dosages higher than 132 kGy acted negatively upon the values of indentation hardness. The application of a 198 kGy radiation dosage led to the decrease of this value compared to that of the unaltered material, and caused degradation of the surface layer. 

The virgin HDPE indentation hardness was 42 MPa. Exposure of this HDPE to beta radiation led to an observable increase in indentation hardness. The highest measured indentation hardness was 66 MPa for test samples irradiated with 198 kGy. The difference between the indentation hardness values for the virgin and irradiated HDPE samples (198 kGy) was 57%. 

Figure 8 shows that the indentation hardness of the tested polyolefin strongly affected the crosslinking process. Decreases in indentation hardness values where test subjects were exposed to radiation dosages higher than 132 kGy for LDPE probably caused material degradation due to the irradiation process.

In addition, the indentation modulus (*E_IT_*) is another material parameter that can be ascertained using the DSI test method (Figure 9). The indentation modulus is determined by using the slope of a tangent curve in order to calculate the indentation hardness (*H_IT_*).

The indentation moduli of the studied polymers were positively affected by radiation crosslinking. The HDPE displayed a gradual increase in indentation modulus throughout the entire range of the applied radiation dosages. The highest indentation modulus was measured in test samples irradiated with a radiation dosage of 198 kGy, namely 1.8 GPa. The indentation modulus rose by 38% in comparison with the virgin material after radiation exposure.

LDPE, while displaying positive results for the indentation modulus, also showed some negative effects for radiation values higher than 132 kGy. The virgin LDPE indentation modulus was 0.19 GPa and the test samples’ indentation modulus when irradiated with a dosage of 132 kGy was 0.23 GPa, representing a 21% increase in comparison to the unaltered material. The higher dosages led to a decrease in the measured properties. Where dosages were higher than 132 kGy, the material exhibited decreases in the indentation modulus values, all the way down to the level of the virgin material.

Another polymer surface layer parameter that is important when the material is exposed to constant stress is indentation creep—this is a situation that is common in technical practice. If the indentation depth is measured during the application of constant stress, the relative indentation depth can be calculated, which corresponds to the material creep value. As can be seen from the results, the test sample irradiation had a positive effect on the material creep properties (Figure 10). The results of this creep were aligned with the indentation hardness and indentation modulus results. 

Measurements of the surface layer’s nano-mechanical properties showed that the application of varying beta radiation dosages caused significant changes in the tested LDPE and HDPE sampels. The irradiation process enhanced the tested materials’ mutual properties, which might enable the development of commodity and construction polymers with the properties of more expensive materials. Further investigation subsequently confirmed these structural and morphological test results.

### 3.2. Structural Properties

#### 3.2.1. Gel Content

The gel content test measures the nonfiltered phase or gel volumes’ specific materials in line with the EN ISO 579 standard. Figure 11 show the determination of the gel content in the selected polymers against the applied radiation dosage. 

Figure 8 and Figure 9 show that the nano-mechanical LDPE and HDPE surface layer properties were affected by the radiation. Enhancement occurred even at the lowest radiation level of 33 kGy. These results disagreed with the gel test, which indicated zero content when exposed to the lowest radiation doses. However, this finding also supported the claim that the lowest amounts of radiation induced micro-gel creation, which the gel test is often unable to detect, while also having a significant effect on mechanical properties. The highest nano-mechanical LDPE property values were found in test samples irradiated with a 132 kGy dosage, in which gel content increased in comparison to the unaltered material. Moreover, the gel content increased gradually with higher radiation doses.

The maximum measured gel content value for an HDPE test sample was for the sample irradiated with a dose of 198 kGy. These findings agreed with the nano-mechanical results measuring similarly altered materials’ highest indentation hardness values. The results of the mechanical tests for the LDPE were not completely confirmed by the gel test. The highest modulus and hardness were measured in materials irradiated by 132 kGy, while the materials exposed to higher radiation levels displayed a decreasing trend in the aforementioned properties. This decline could have been caused by degradation processes, which were not factored into the gel content test.

#### 3.2.2. Infrared Spectroscopy

Figure 12 and Figure 13 illustrate the HDPE and LDPE infrared spectra, respectively. The HDPE spectra showed decreases in the intensity of bands ascribed to aliphatic CH bonds (e.g., negative bands at 3000–2800, ~1450, and ~1370 cm^−1^) and increases in the intensity of corresponding oxygenated functional group bands at 1595 and 3360 cm^−1^. Furthermore, the LDPE spectra showed increases in the bandwidth of oxygen functional groups at 1719 and 3405 cm^−1^.

Based on the infrared spectra, the conclusion was drawn that irradiation caused CH bonds (–CH_2_–) in HDPE and LDPE to oxidize, and carbonyl (–CO–) and hydroxyl (–OH) groups formed as a result. Figure 14 and Figure 15 (for hydroxyl and carbonyl groups, respectively) present the increase in the relative numbers of oxygen functional groups. 

Figure 14 and Figure 15 show that the relative numbers of hydroxyl and carbonyl groups in all tested materials depended on the absorbed radiation dose. Non-irradiated materials showed the lowest relative numbers of hydroxyl groups. The relative number of oxygen-containing groups increased in line with radiation dose increase. The radiation dose of 165 kGy applied to HDPE and LDPE produced the highest increase in the relative number of oxygen-containing groups.

The changes in the LDPE and HDPE surface layer properties were probably due to oxidation that occurred during and after beta irradiation. Oxidation is one of the secondary reactions that occurs when ionizing radiation interacts with a polymer [30], and results in the formation of the above-mentioned carbonyl and hydroxyl functional groups. Atmospheric oxygen diffusion throughout the polymer bulk probably controls the kinetic processes that caused the degradation of these products.

Alkyl and allyl radicals reacted with oxygen molecules to form carbonyl groups on the surfaces of samples, while within the samples, alkyl radicals led to crosslinking and allyl radicals reacted with oxygen to form hydro-peroxide groups. These findings agreed with the results reported by Murray [31] and Hama [32]. In addition to oxidation, which affects polymer surface properties, post-irradiation oxidation also influences polymer properties. Carpentieri [33] and Costa [34] mentioned that post-irradiation oxidation is one of the factors that influence polymer crystallinity and crystalline lamella size. This could explain to a certain extent some of the differences in the relative numbers of oxygenated HDPE and LDPE functional groups (Figure 14 and Figure 15, respectively).

LDPE showed a lower relative number of oxygenated functional groups that presented lower crystallinity than HDPE. This is probably attributable to differences in reactivity between macro-alkyl radicals that were able to form in the polymers’ amorphous and crystalline phases during irradiation. 

Therefore, the conclusion was drawn that, for LDPE, post-irradiation oxidation was very low, and was achieved in a relatively short time. The LDPE lamellae were very thin when compared to HDPE lamellae; therefore, the migration time of the macro-alkyl radicals from the crystalline to the amorphous phase was very short. Conversely, the HDPE oxidation rate was higher and continued for a relatively long time after the irradiation process. The wider HDPE crystalline lamellae captured the macro-allyl radicals, and thus their migration time to the amorphous phase was longer. Hence, migration times of radicals from crystalline to amorphous phases appeared to be one of the key factors governing the oxidation process.

#### 3.2.3. Raman Spectroscopy

The DSI method confirmed the tested polymer groups’ results and nano-mechanical property changes based on additional measurements, the goal of which was to describe these beta-irradiated polymers according to their structural changes and morphology. A single material sample from each group was chosen as a group representative. 

The LDPE and HDPE Raman spectroscopy test samples displayed only marginal changes to their structure in comparison to the virgin materials (Figure 16 and Figure 17).

#### 3.2.4. X-ray Diffraction

X-ray diffraction of the HDPE showed that the highest crystalline phase content was present in the virgin material, which also exhibited the lowest nano-mechanical surface-layer property values. In addition, we observed a decrease in line with the crystalline phase content with increasing radiation intensity. The lowest values were recorded for the test sample irradiated by 165 kGy of radiation, as can be seen in Figure 18 and Figure 19. These findings confirmed the changes in the measured properties, i.e., indentation hardness, indentation modulus, and indentation creep. Figure 8 and Figure 9 depict changes to the indentation hardness and indentation modulus.

Figure 19 and Figure 20 display the X-ray diffraction used to determine the LDPE crystalline content. The highest values found were in test subjects irradiated with radiation dosages of 66, 99, and 132 kGy, while virgin test samples showed the lowest values. These results aligned with the surface layer nano-hardness and nano-rigidity measurements. Higher radiation dosages resulted in decreasing content of the crystalline phase.

## 4. Conclusions

The ionizing beta radiation results indicated a positive effect on the tested materials’ nano-mechanical properties. The LDPE and HDPE property changes differed with increasing radiation dosages.

Polyolefin polymer representative materials are often used in technical practice. HDPE and LDPE were the representative polymers. Electron radiation induced the creation of a 3D network within the selected polymers’ structures, and thus significantly altered the polymers’ final behavior.

The state-of-the-art DSI method was used to measure the tested polymers’ surface layer nano-mechanical properties. This method enabled observation of polymer behavior during testing tip penetration into the material, with subsequent evaluation of the tested samples’ mechanical properties. HDPE exposed to 198 kGy of radiation demonstrated the greatest changes in surface layer mechanical properties, e.g., indentation hardness, in comparison to the virgin material. The resulting indentation hardness was 57% higher than that of the unaltered material.

The highest LDPE indentation hardness value, in the sample irradiated by 132 kGy, was 26 MP, or 24% higher than the virgin polymer. The indentation modulus and indentation creep results showed similar tendencies. In comparison with the virgin material, the greatest changes to indentation modulus and indentation creep were measured in HDPE irradiated by 198 kGy (38% and 8%, respectively), and LDPE irradiated by 132 kGy (21% and 11%, respectively).

HDPE exhibited a decrease in nano-mechanical surface layer property values when exposed to higher radiation doses, perhaps caused by surface layer degradation. Thus, the final radiation dose required careful selection. In some cases, lower radiation dosages led to similar degrees of modification as higher radiation doses.

Examinations of the polymer structures supported the LDPE and HDPE surface layer nano-mechanical property results from the gel content test, i.e., the creation of a 3D network within the polymer structure, and the X-ray diffraction and Raman spectroscopy proved and confirmed these changes by measuring the crystalline degree.

This study’s aim was to explore the possibility of using electron radiation to modify some commonly used polymers (polyolefins). This could lead to an enhancement of the desired properties for a fraction of the cost that would be required if taking the construction polymers route. It is imperative to note that higher radiation amounts do not always equal a better result. The optimal radiation dosage must be investigated individually for each material and application. The measured results will provide a starting point leading to more detailed study. The effectiveness of radiation modification is a key area that particularly requires further study.

## Figures and Tables

**Figure 1 materials-13-00929-f001:**
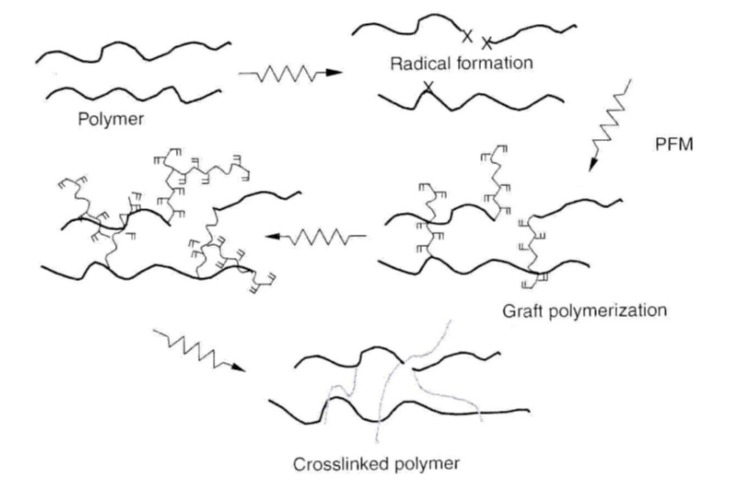
Schematic representation of irradiated polymer crosslinking mechanism [2].

**Figure 2 materials-13-00929-f002:**
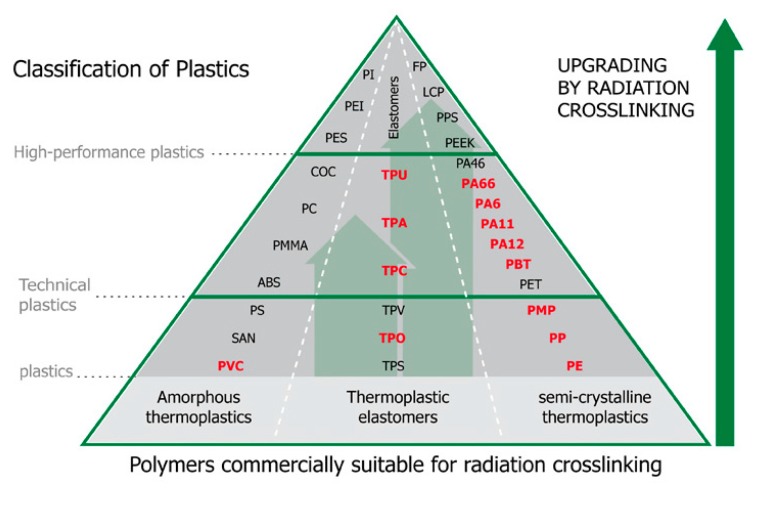
Influence of radiation crosslinking on plastics [24,25].

**Figure 3 materials-13-00929-f003:**
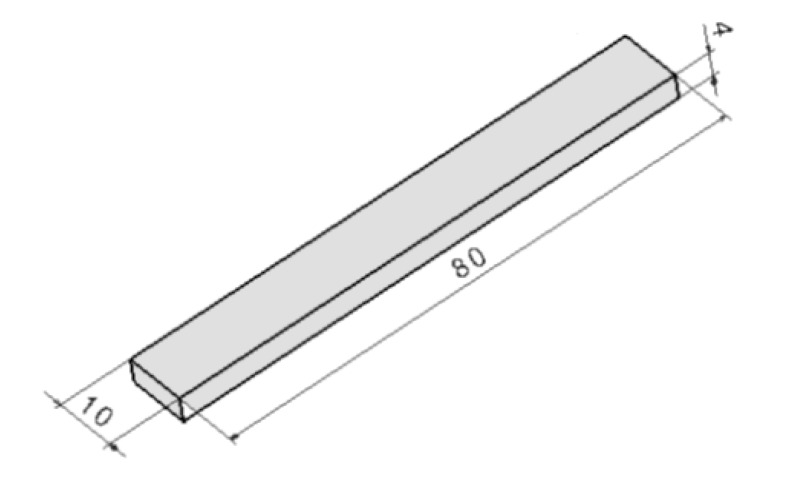
Test sample dimensions (mm).

**Figure 4 materials-13-00929-f004:**
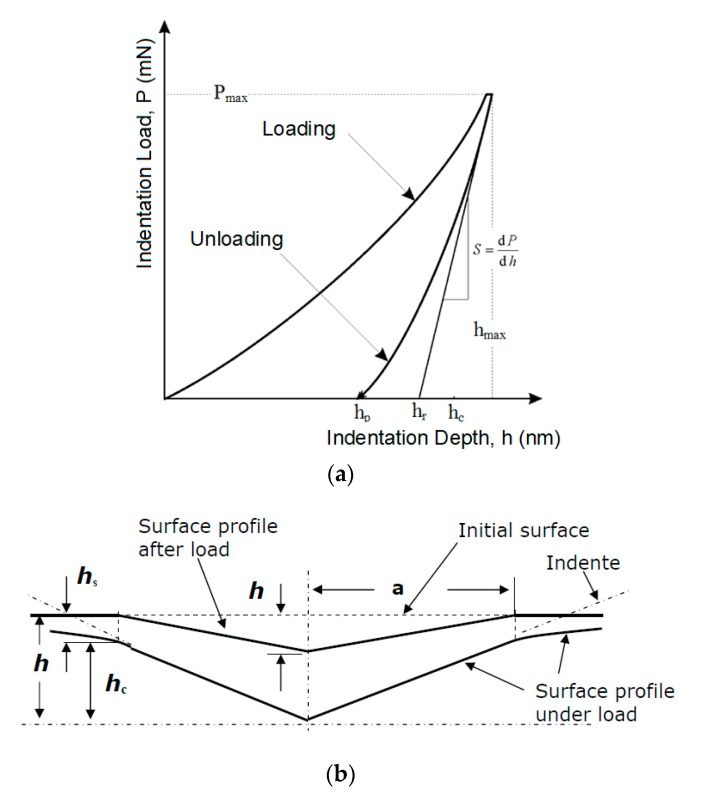
(**a**) Load–displacement curve, showing the values used in the Oliver and Pharr method [26]; (**b**) cross-section of an indentation [26].

**Figure 5 materials-13-00929-f005:**
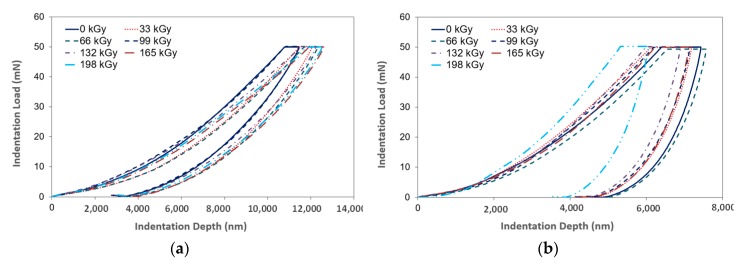
Indentation curve (indentation load vs. indentation depth): (**a**) LDPE; (**b**) HDPE.

**Figure 6 materials-13-00929-f006:**
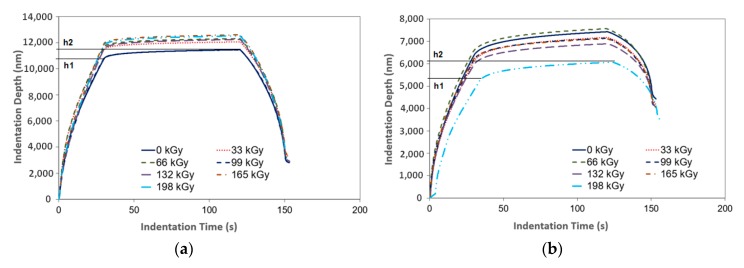
Indentation creep (indentation depth vs. indentation time): (**a**) LDPE; (**b**) HDPE

**Figure 7 materials-13-00929-f007:**
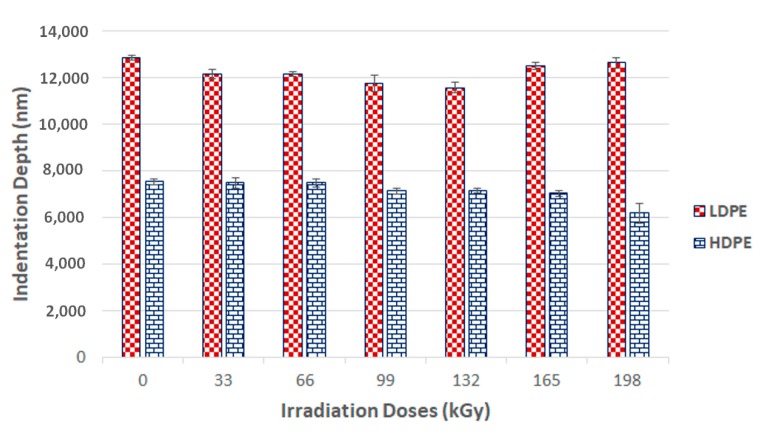
Indentation depth (*h*_max_) of irradiated LDPE and HDPE; the vertical line drawn at the top of the column represents the standard deviation.

**Figure 8 materials-13-00929-f008:**
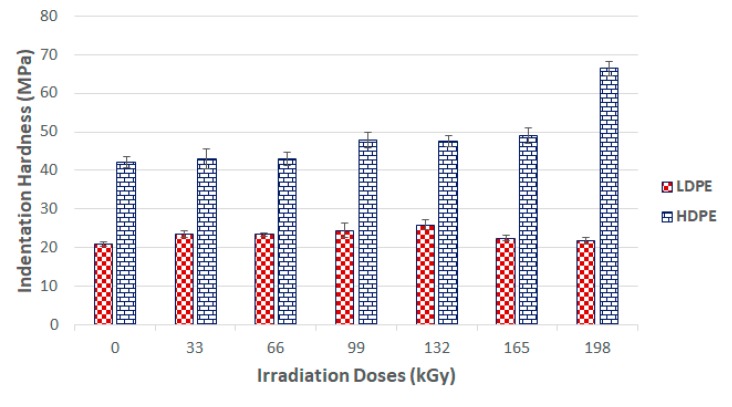
Indentation hardness (*H*_IT_) of irradiated LDPE and HDPE; the vertical line drawn at the top of the column represents the standard deviation.

**Figure 9 materials-13-00929-f009:**
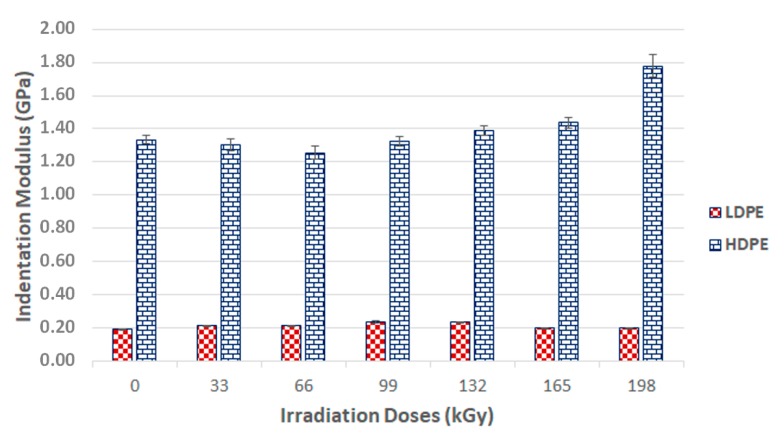
Indentation modulus (*E_IT_*) of irradiated LDPE and HDPE; the vertical line drawn at the top of the column represents the standard deviation.

**Figure 10 materials-13-00929-f010:**
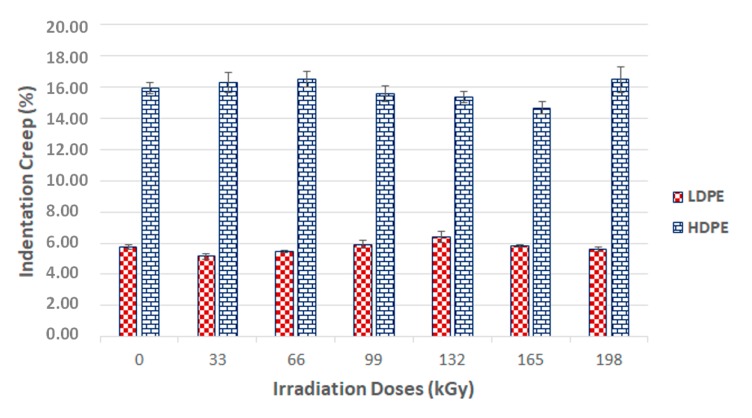
Indentation creep (*C_IT_*) of irradiated LDPE and HDPE; the vertical line drawn at the top of the column represents the standard deviation.

**Figure 11 materials-13-00929-f011:**
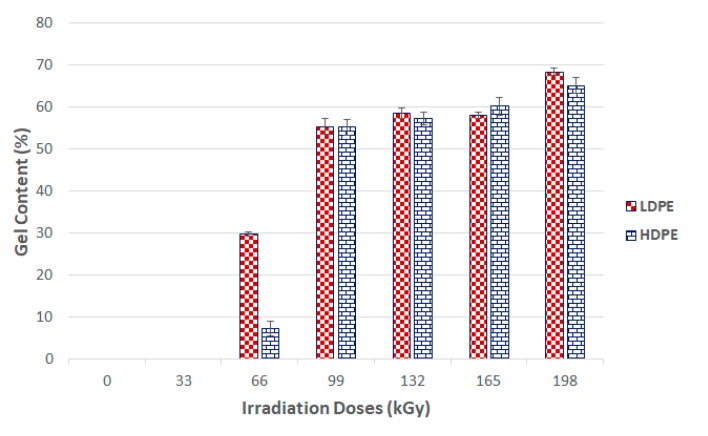
Gel content of irradiated LDPE and HDPE; the vertical line drawn at the top of the column represents the standard deviation.

**Figure 12 materials-13-00929-f012:**
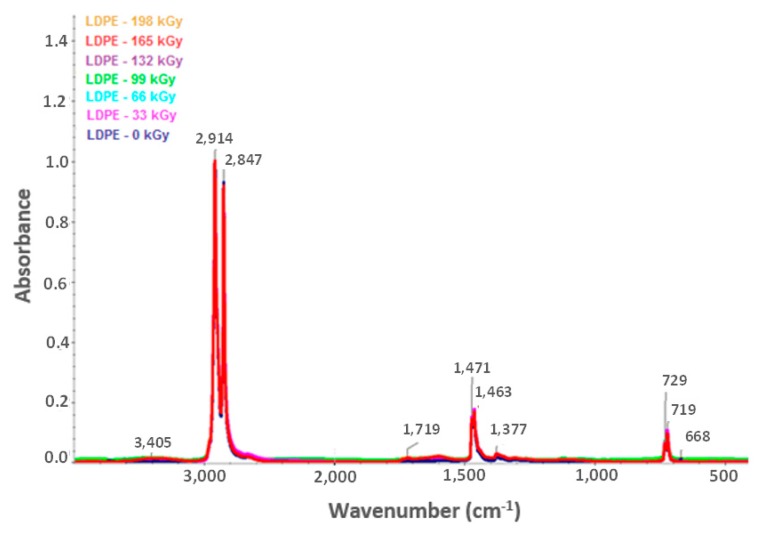
Infrared spectra of irradiated LDPE [29].

**Figure 13 materials-13-00929-f013:**
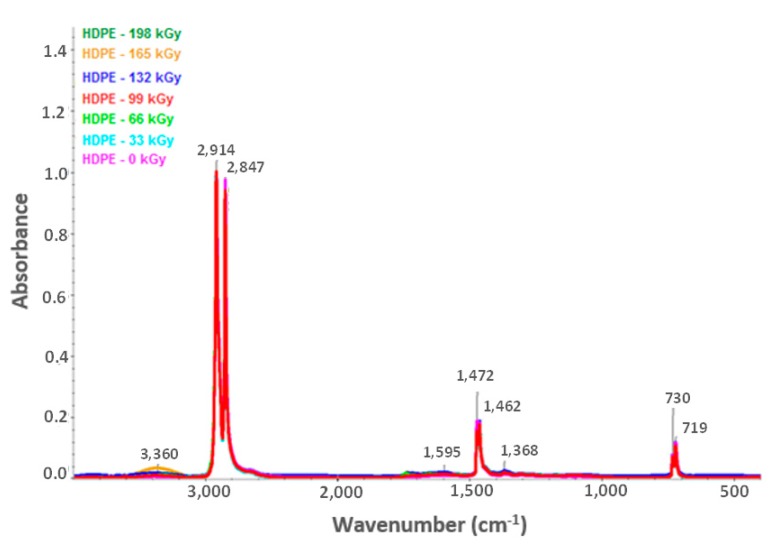
Infrared spectra of irradiated HDPE [29].

**Figure 14 materials-13-00929-f014:**
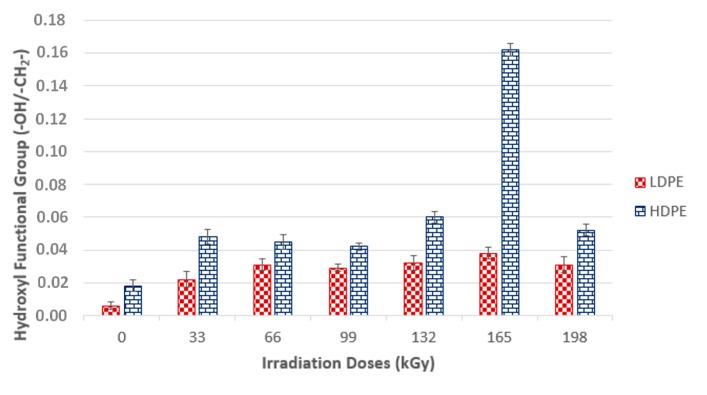
Relative number of hydroxyl functional groups and radiation dose relationships; the vertical line drawn at the top of the column represents the standard deviation.

**Figure 15 materials-13-00929-f015:**
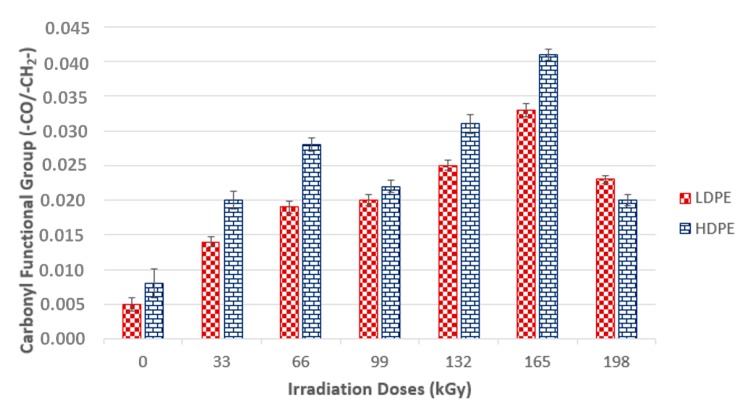
Relative number of carbonyl functional group and radiation dose relationship; the vertical line drawn at the top of the column represents the standard deviation.

**Figure 16 materials-13-00929-f016:**
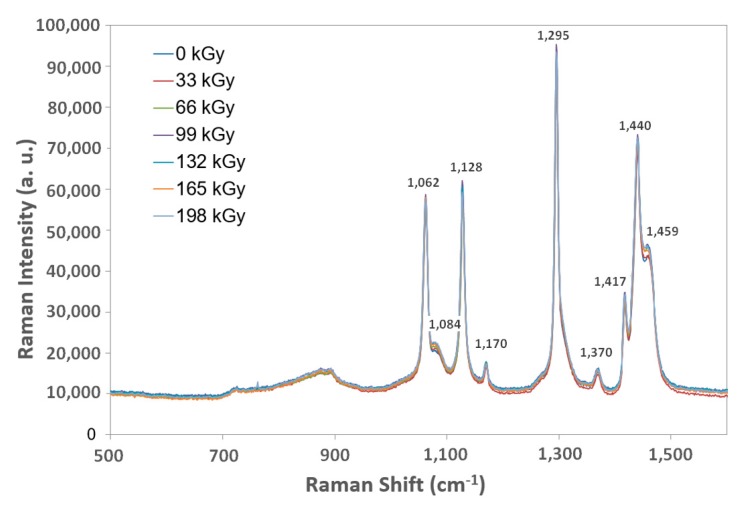
Raman spectroscopy of irradiated LDPE.

**Figure 17 materials-13-00929-f017:**
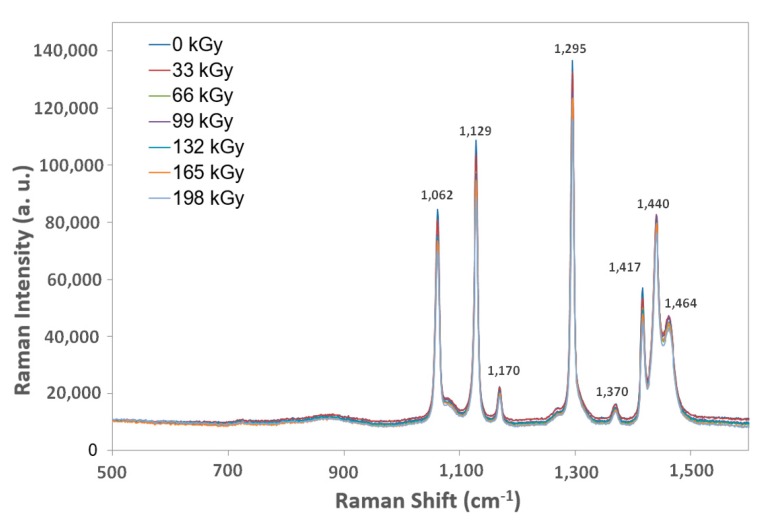
Raman spectroscopy of irradiated HDPE.

**Figure 18 materials-13-00929-f018:**
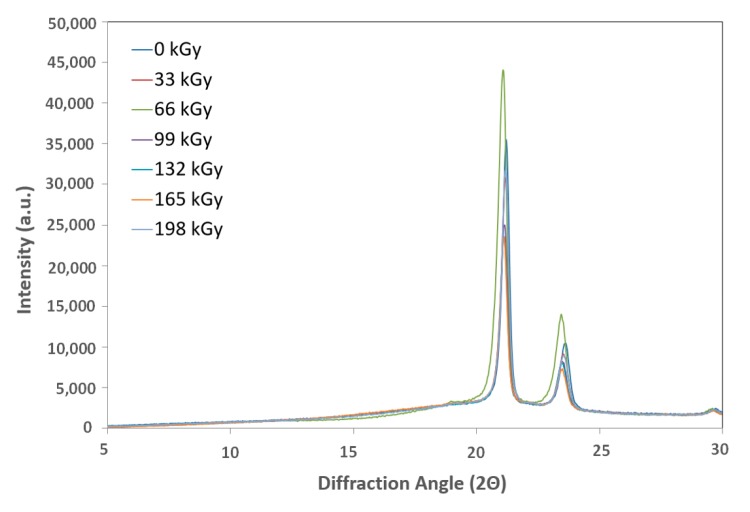
X-ray diffraction of irradiated HDPE.

**Figure 19 materials-13-00929-f019:**
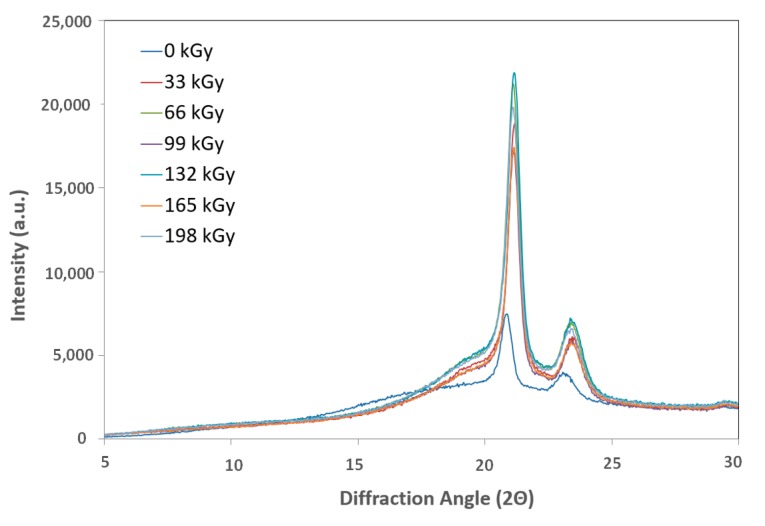
X-ray diffraction of irradiated LDPE.

**Figure 20 materials-13-00929-f020:**
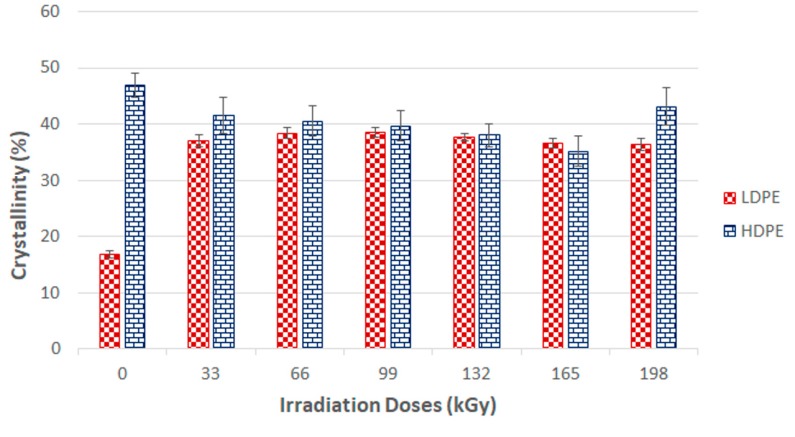
X-ray diffraction crystallinity of irradiated LDPE and HDPE; the vertical line drawn at the top of the column represents the standard deviation.

**Table 1 materials-13-00929-t001:** Tested polymers.

Type of Polymers	Trade Name	Company
Low-density polyethylene	LDPE (LDPE DOW 780E)	DOW (Midland, MI, USA)
High-density polyethylene	HDPE (HDPE DOW 25055 E)	DOW (Midland, MI, USA)

**Table 2 materials-13-00929-t002:** Process parameters.

Process Parameter	LDPE	HDPE
Injection rate (mm/s)	50	60
Injection pressure (MPa)	60	80
Cooling time (s)	30	35
Mould temperature (°C)	40	40
Holding time (s)	30	25
Barrel temperature—Zone 1 (°C)	140	150
Barrel temperature—Zone 2 (°C)	150	160
Barrel temperature—Zone 3 (°C)	160	180
Barrel temperature—Zone 4 (°C)	180	190

**Table 3 materials-13-00929-t003:** Process parameters.

Parameters	Unit	Value
Maximum load	mN	50
Load/unload speed	mN/min	100
Holding time (*H_IT_*, *E_IT_*)	s	90
Holding time (*C_IT_*)	s	21,600

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
