# Peer review of "Nano-Mechanical Properties of Surface Layers of Polyethylene Modified by Irradiation"

_materials, 2020, doi:10.3390/ma13040929_

Round 1

Reviewer 1 Report

Title: Nano-mechanical properties of surface layers of polyethylene modified by irradiation

Authors: Martin Ovsik and colleagues

Overall assessment:

The authors investigate the effects of irradiation on various properties of low- and high-density polyethylene samples. The topic adopted in this manuscript may be interesting for some researchers. Nevertheless, the authors might not conduct several objective readings prior to the submission; therefore, there are many casualties in the descriptions. In the next occasion, the authors must improve the manuscript by careful and objective readings into that many readers, including undergraduate students, can understand the essence of the study easily and appropriately.

Specific comments:

Table 1: “High density polyethylene” is mistyped as “Hight density polyethylene”. Additionally, the trade names “LDPE DOW 780E”and “DOW HDPE 25055 E” are really correct, aren’t they? “DOW HDPE 25055 E” is the mistype of “HDPE DOW 25055 E”, isn’t it? The authors should check the trade names of listed in this table carefully.

The samples with dumbbell and bar shapes shown in Fig. 2 seem to be used for the tension and Charpy tests, respectively. However, it is difficult to understand how to use these specimens because the corresponding descriptions are too terse. It is not adequate to list the standards alone, and more detailed explanations are required.

The unit is missing in Fig. 2. Even if the unit is evident for the readers, it must be denoted in the figure or the caption.

It is difficult to understand what “Zone” in Table 2 represents.

Figure 4: It is meaningless to show a photograph of the tester alone. Where are the sample and indenter? It is necessary to address the explanations in the photograph. Otherwise, a diagram of the nano-indentation test, including the shape of the penetrating body, should be demonstrated instead of the photograph.

The Poisson’s ratio is defined as in the line 139, but it is similar to s in Eq. (2), isn’t it?

There are no descriptions on the methods how to obtain the Ap value in Eq. (1) and E* and s values in Eq. (2). Since the indentation hardness HIT and indentation modulus EIT calculated from the Ap, E*, and s values are used for the discussion in this manuscript, it is necessary to demonstrate the methods.

It is difficult to understand how to obtain the values of the indentation depth shown in Figs. 5-7 using the values of h1 and h2, which are also defined as the “indentation depth” in the lines 141 and 142.

The captions of Figs. 5-20 are extremely too terse. From the captions, it is often difficult to understand what the authors would like to insist on using these figures. Additionally, it is difficult to understand what the vertical line drawn at the top of the column represents. Although the line may represent the standard deviations, the definition must be denoted in the caption.

From these reasons, the authors must elaborate the captions thoroughly.

It is difficult to understand why the authors demonstrate Table 4 and Fig. 11, which represent the similar issues.

I cannot find any descriptions on the results of tension and Charpy tests, the samples of which may be demonstrated in Fig. 2.

Recommendation

 Thorough revisions are required and the authors must prepare the revised version as well as the point-by-point response to the comments described above. The casual descriptions often prevent the readers from precise understandings of the essence of this study; therefore, the authors must improve the manuscript by careful and objective readings. I’d like to receive a revised version in due course.

Author Response

Thank you very much for your comments regarding our paper. They are quite useful for us and help us improve the quality of the prepared paper. We gratefully accept your comments and the changes according to your recommendation were included in the corrected version of the paper. I hope the correction will meet your requirements and you will accept it. Concerning English language, the language proof of the text was done by native speaker.

An explanation and supplement of your comments is attached.

Reviewer 2 Report

In this work, authors study quite exhaustively the effect of different irradiation doses on several nano-mechanical properties of two common polymers, LDPE and HDPE, and relate those properties with the formation of 3D polymer networks. I find the work interesting, valuable for researches on the topic and useful to develop technological applications of the tested materials. The work is essentially correct and well done. On the other hand, the paper is clearly written and the conclusions are well supported by the results. In my opinion, it can be published in Materials as it is.

Author Response

Thank you very much for your comments regarding our paper. They are quite useful for us and help us improve the quality of the prepared paper. Concerning English language, the language proof of the text was done by native speaker.

Reviewer 3 Report

The authors presented a study of the effect of irradiation on the nano-mechanical properties of surface layers of polyethylene. Several experimental techniques were used. I have the following comments for authors to consider.

1, the abstract is too long! it looks and reads like an introduction. Please rewrite the abstract.

2, the references are badly written. There are many locations where the authors should show a reference. For example, oliver-pharr method (cite their paper). Line 55: other research? what are they?

3, Line 77 to 80. The aim of the paper.....Herethe main difference of this paper compared to literature should be highlighted, I could not see from the introduction what exactly is new in this paper. All of the experimental techniques were already there, the authors just did several tests, so what's new?

4, line 110. All unit in figure caption

5, figure 3, any specific reason to use two types of arrow in the figure? And I do not see any reason Why The First Letter Of Each Word Should Be In Capital!

6, line 131, a three sided diamond pyramid? for indentation community, people would just say: Berkovich indentation

7, table 3. I believe the loading rate is quite important here for these materials. Why you choose this loading velocity? Did you see any dependence on the loading velocity?

8, equation 4. dot product is missing.

9, line 219. on stress, it is not stress,it is force.

10, figure 6. It is not a typical creep curve since force is not really a constant.The other thing I do not get is the creep really depends on how long you hold the force. especially for Figure 6b, the slope between depth and time is pretty obvious.

11. Line 264. In ideal cases, the indentation modulus carries the same weight
as the elastic (Young) modulus. I do not really understand what the authors tend to say.

12, figure 11. I do not really like the result since it is not consistent with the mechanical testing in the way that the higher gel content does not correspond to the higher increase of the hardness. So this is also the biggest drawback of this paper. For sure radiation changes the material property, however, if one cannot completely understand when and how radiation has the highest change to the material strength, I do not see any scientific importance of doing so many different tests while the only conclusion is that: they change the material property. I can even claim this without doing so many experiments. 

13, line 390. 20As ????

14, There are several English problems, for example, line 402, changes differed with....line 405, altering..

Author Response

(The authors gave the same response as above.)

Reviewer 4 Report

The purpose of this paper was to describe the influence of a wide range of ionizing beta radiation upon changes in surface layer mechanical properties and structural modifications of the selected types of polymers. Paper requires some adjustments.

Authors should answer the following questions and make changes in the text: 

Keywords

-          The word spectroscopy should be added.  

Introduction

-          The review of reference sources has been carried out correctly.

-          Authors can think about addition their own papers connected with the polymers to the introduction. Some information from the chapter Materials and Methods should be moved.   

Materials and Methods

-          The Table 1 is not necessary in the paper. Instead Authors should give more information connected with the properties of LDPE and HDPE used for the investigations. Why Authors used commercial products from DOW the Company?

-          How many samples were used for the experiments?

-          What about conditioning time of samples before experiments?

-          The information given on the Figure 3 in the Introduction chapter should be given.

-          Sentences given in the 2.5. chapter are not necessary: “… The chemical linking of two polymer functional groups produces linear polymers. Furthermore, if one of the groups is multifunctional, swelling and elongation with branching is observable. This leads to the creation of an infinite 3D network, i.e. gel.  In all cases, there are cross-linking actions that happen sequentially: Dimensions and poly-dispersion grow in the first phase; As the reaction reaches gel point, the molar weight increases and the gel starts to appear in the system; The system includes two parts – on the one hand, an infinite structure called ‘gel’; on the other,  a cluster of molecules with finite size - called ‘sol’. Extraction of molecules leads to its separation from the gel.; It is not possible to extract the complete network, by-products and micro-gel. The gel is insolvable - and if exposed to a solvent, it experiences imbibition.; Finally, the sol content - as well as its molar weight and poly-dispersity, decreases as the reaction continues.; The gel network creates active elastic chains. These chains carry the applied stress, thus determining the value of the gel elastic modulus and its imbibition equilibrium degree…”. They are not direct information connected with the experiments! Authors should remove it! 

Results

-          Authors correctly analyzed results of the indentation estimation. There is an interesting tool used for the analysis different polymers.-          The information given in the Table 4 and Figure 11 are the same! Authors should decide which presentation is better: Table or Figure?

-          Some Figures need description of the X-axis or Y-axis (e.g. 12-15, 18-19). 

Conclusions

-          The first sentence “…The goal was to describe varying ionizing beta radiation dosages and their influence on changes to tested polymers surface layer nano-mechanical properties. …” is not a conclusion. Authors should remove them. 

References

-          The selection of the references was made correctly. 

Paper can be published after minor changes.

Author Response

(The authors gave the same response as above.)

Round 2

Reviewer 1 Report

The revisions are adequately conducted; therefore, I can recommend the revised version to be published in the journal as it is.